# Exogenous Ceramide Serves as a Precursor to Endogenous Ceramide Synthesis and as a Modulator of Keratinocyte Differentiation

**DOI:** 10.3390/cells11111742

**Published:** 2022-05-25

**Authors:** Kyong-Oh Shin, Hisashi Mihara, Kenya Ishida, Yoshikazu Uchida, Kyungho Park

**Affiliations:** 1Department of Food Science & Nutrition, Convergence Program of Material Science for Medicine and Pharmaceutics, Hallym University, Chuncheon 31151, Korea; 0194768809@hanmail.net; 2The Korean Institute of Nutrition, Hallym University, Chuncheon 31151, Korea; 3LaSS Lipid Institute (LLI), LaSS Inc., Chuncheon 31151, Korea; 4Takasago International Company, Hiratsuka 259-1207, Japan; hisashi_mihara@takasago.com (H.M.); kenya_ishida@takasago.com (K.I.); 5Veterans Affairs Medical Center, Department of Dermatology, School of Medicine, Northern California Institute for Research and Education, University of California, San Francisco, CA 94158, USA

**Keywords:** ceramide, barrier, differentiation, keratinocyte, skin

## Abstract

Since ceramide is a key epidermal barrier constituent and its deficiency causes barrier-compromised skin, several molecular types of ceramides are formulated in commercial topical agents to improve barrier function. Topical ceramide localizes on the skin surface and in the stratum corneum, but certain amounts of ceramide penetrate the stratum granulosum, becoming precursors to endogenous ceramide synthesis following molecular modification. Moreover, exogenous ceramide as a lipid mediator could modulate keratinocyte proliferation/differentiation. We here investigated the biological roles of exogenous NP (non-hydroxy ceramide containing 4-hydroxy dihydrosphingosine) and NDS (non-hydroxy ceramide containing dihydrosphingosine), both widely used as topical ceramide agents, in differentiated-cultured human keratinocytes. NDS, but not NP, becomes a precursor for diverse ceramide species that are required for a vital permeability barrier. Loricrin (late differentiation marker) production is increased in keratinocytes treated with both NDS and NP vs. control, while bigger increases in involucrin (an early differentiation marker) synthesis were observed in keratinocytes treated with NDS vs. NP and control. NDS increases levels of a key antimicrobial peptide (an innate immune component), cathelicidin antimicrobial peptide (CAMP/LL-37), that is upregulated by a ceramide metabolite, sphingosine-1-phosphate. Our studies demonstrate that NDS could be a multi-potent ceramide species, forming heterogenous ceramide molecules and a lipid mediator to enhance differentiation and innate immunity.

## 1. Introduction

An epidermal permeability barrier that prevents excess water evaporation, loss of ions, and other small molecules from the skin is essential for terrestrial mammalian survival. This permeability barrier is also responsible for preventing the penetration of exogenous molecules and the invasion of microbes. Lipid-dominant, extracellular lamellar membrane structures in the stratum corneum (SC) layer of the epidermis are largely responsible for permeability barrier function; and cholesterol, free fatty acid (FA), and ceramide (Cer) are the major constituents of the epidermal permeability barrier [1]. The Cer in the SC is made up of heterogeneous molecular species (21 molecular groups are identified in humans) [2], and this molecular heterogeneity has been identified only in the differentiated epidermal layers of terrestrial mammals. Not only the bulk but also a certain molecular ratio of each Cer species, is important to form vital barrier structures [3,4]. Deficiencies in ω-O-acylCer (acylCer) and non-hydroxy acyl 4-hydroxydihydrosphingosine (phytosphingosine [P]) (NP), as well as changes in the composition of amide-linked FA species, are found in atopic dermatitis patient skin [5,6,7,8,9]. Additionally, Cer profile changes are found in psoriatic skin [10,11,12] and xerosis [13]. Since the topical application of Cer improves epidermal permeability function [14,15,16], Cer is formulated as an active ingredient in topical products to attenuate disease symptoms of skin diseases associated with compromised permeability barrier function. Cer species such as NP, 2-hydroxy acylP (AP), non-hydroxy acyldihydrosphingosine (NDS), 2-hydroxy acyldihydrosphingosine (ADS), acylCer containing P (EOP), and structure mimic pseudoceramides are used in skincare products to improve barrier function [17]. AcylCers are essential in SC lamellae formation, while the ability to form long periodicity lamellar phases (LPP) required for vital barrier function varies with different EOS (acylCer containing sphingosine [S]), EOP, EOH (acylCer containing 6-hydroxyS[H]), and EODS (acylCer containing [DS) [18,19]. Additionally, levels of NS and Cer containing shorter amide-linked FA are increased in atopic dermatitis [8]. Hence, the proper selection/amount of ceramide species for a topical formulation is very important for the improvement/correction of skin barrier structure and function. Note that the presence of Cer containing 4,14-sphingadiene (SD) and a ß-hydroxy acyl sphingoid base was identified in 2020 [2], but neither physiological nor pathological roles of these Cer species in SC have been explored.

Topically applied Cer could improve barrier function in three possible ways: (1) Cer in an appropriate formulation constructs (lamellar) liquid crystalline and/or gel structures [20]; (2) Cer is likely incorporated into lamellar bilayer structures in the SC to repair incompetent bilayer structures [21]; and/or (3) Cer penetrates into nucleated layers of the epidermis [22,23] and becomes a precursor to endogenous Cer synthesis following hydrolysis to the sphingoid base and FA (see Cer synthetic pathway in Figure 1A). In addition, Cer absorbed into nucleated layers of the epidermis could suppress KC proliferation and promote differentiation [24]. Because Cer production is increased during KC differentiation, the stimulation of KC differentiation can increase *de novo* Cer production, thereby contributing to barrier formation.

Stimulation of endogenous Cer production is a therapeutic strategy used to improve epidermal permeability barrier integrity. Yet, compounds to specifically promote Cer production have not yet been developed. Since increases in precursor levels increase Cer production [26], the supplementation of Cer and its precursor is an achievable approach to elevate epidermal Cer levels. In theory, dihydroCer (NDS) is converted into diverse types of Cer, while NS can only be a precursor of Cer species containing S (Figure 1B). Similarly, NP can only be a precursor of Cer species containing P (Figure 1B). However, the fate of exogenous Cer in the epidermis has not yet been elucidated, nor has the role(s) of exogenous Cer in the modulation of KC proliferation/differentiation.

The amount of exogenous Cer absorption into the epidermis is dependent upon the barrier integrity, and the penetration of exogenous Cer is relatively low in barrier-competent (normal) skin [27]. Recent studies have aimed to increase Cer absorption to increase the efficacy of Cer to further improve barrier function [28,29], suggesting that exogenous Cer can be absorbed into nucleated layers of the normal epidermis. Hence, elucidating the fate of exogenous Cer in the epidermis is important to understand whether exogenous Cer modulates KC differentiation, as well as contributes to barrier formation.

Here, we aim to define the fates of exogenous Cer, the contribution of exogenous Cer to increase Cer levels in differentiated KC, and the effect of exogenous Cer on KC differentiation. As above, several types of Cer species are used in topical agents. Because NDS and NP are widely used as skincare ingredients, we employed these Cer in this study.

## 2. Materials and Methods

### 2.1. C17 Cer Synthesis

Both N-stearoyl-D-erythro-dihydrosphingosine (C17 base) (17NDS) and N-stearoyl-D-erythro-4-hydroxydihydrosphingosine (C17 base) (17NP) were synthesized, according to the following procedure: D-erythro-dihydrosphingosine(C17 base) and D-ribo-phytosphingosine (C17 base) (Avanti Polar Lipid) are dissolved in ethanol at a concentration of 10 mM and then mixed into 50 mL tetrahydrofuran, containing stearoyl chloride (final concentration 0.2 mmol) and triethylamine (final concentration 0.2 mmol) and incubated for 18 h at 70 °C. Reacted mixtures were added to 50 mL water and 100 mL of ethylene acetate to extract organic solvent fractions, which contain 17NDS or 17NP. 17NDS or 17NP was purified by silica gel (particle size: 0.040–0.063 mm, pore size: 60 Å, Sigma Aldrich, St. Louis, MO, USA) column chromatography (hexane:ethylene acetate 17:3, *v*/*v*).

### 2.2. Cell Culture

Human primary keratinocytes (KCs) from Life Technologies (Carlsbad, CA, USA) were maintained in serum-free KC growth medium containing 0.07 mM Ca^2+^. Cells at 60–70% confluence were further cultured in a differentiation-inducing medium in two consecutive steps as described (Appendix A). First, KC were incubated in DMEM and Ham’s F-12 (2:1, *v*/*v*) containing 1.2 mM calcium, 10% FBS, and 10 µg/mL insulin, along with vitamin C (50 µg/mL) for 8 days with Cer, as described previously [30,31]. Cells were then incubated in DMEM containing 1.2 mM calcium, 10% FBS, and 10 µg/mL insulin, along with vitamin C (50 µg/mL) for 2 days incubated with Cer. Synthesized 17NDS or 17NP were dissolved in 95% aqueous ethanol in a sonication bath for 30 min at 40 °C at a concentration of 20 mM prior to each treatment.

### 2.3. Cell Viability Assay

Cell viability or cytotoxicity was determined by the water-soluble tetrazolium salt (WST) method using the Cell Counting Kit-8 (CCK-8, Dojindo, Japan) in accordance with the manufacturer’s instructions.

### 2.4. Lipid Analysis

Total lipids were extracted from cells by the method of Bligh and Dyer, with modification, as described previously [32,33]. Cer and Sphingosine-1-Phosphate (S1P) were quantitated using LC-ESI-MS/MS (API 3200 QTRAP mass), as described previously [32,33,34]. The sphingoid base MS/MS transitions (*m*/*z*) were 286→268 for C17 S and 288→60 for C17 ND as an internal standard and 300→282 for C18 sphingosine and 302→60 for C18 ND, respectively. The sphingoid bases-1-phosphate MS/MS transitions (*m*/*z*) were 366→250 for C17 S1P as an internal standard and 380→264 for C18 S1P and 382→284 for C18 sphinganine-1-phosphate, respectively. The Cer MS/MS transitions (*m*/*z*) were 510→264 for C14Cer, 538→264 for C16-Cer, 552→264 for C17Cer, 566→264 for C18Cer, 594→264 for C20Cer, 648→264 for C24:1Cer, 650→264 for C24Cer, 676→264 for C26:1Cer, and 678→264 for C26Cer, respectively. Data were acquired using Analyst 1.7.1 software (Applied Biosystems, Foster City, CA, USA). Sphingolipid levels are expressed as pmol per mg protein. Protein content was determined using the BCA protein assay method (Pierce, Rockford, IL, USA), using bovine serum albumin as the standard.

### 2.5. Western Blot Analysis

Western blot analysis was performed, as described previously [33]. Briefly, cell lysates, prepared in radioimmunoprecipitation assay buffer (RIPA Lysis and Extraction Buffer), were resolved by electrophoresis on 4–12% Bis-Tris protein gel (Invitrogen, Waltham, MA, USA) under denaturing conditions using SDS. All procedures are conducted following the manufacturer’s instructions. Resultant bands were blotted onto polyvinylidene difluoride membranes, probed with anti-human keratin 10 (Santa Cruz Biotechnology, Dallas, TX, USA), anti-human involucrin Abcam (Cambridge, UK), or anti-human β-actin (Sigma-Aldrich), and detected using enhanced chemiluminescence (Thermo Fisher Scientific, Waltham, MA, USA). The intensity of bands was measured with a LAS-3000 (Fujifilm, Tokyo, Japan).

### 2.6. Quantitative RT-PCR 

mRNA expression was assessed by quantitative RT-PCR (*q*RT-PCR) using SensiFAST™ SYBR^®^ Lo-ROX Kit (Bioline/Meridian, Memphis, TN, USA), as described previously [33]. Briefly, total RNA was isolated from cell lysates using the RNeasy mini kit (Qiagen, Hilden, Germany), followed by the preparation of cDNA using the SensiFAST^TM^ cDNA synthesis kit (Bioline/Meridian). The used primer sets for PCR are listed below. The thermal cycling conditions were 95 °C for 10 min, 95 °C for 15 s, 60 °C for 15 s, and 72 °C for 15 s, repeated 40 times on ABI Prism 7500 (Applied Biosystems). mRNA expression was normalized to levels of GAPDH. Values shown represent the mean (± SD) for three independent assays. The following primer sets were used: CAMP, 5′-CACAGCAGTCACCAGAGGATTG-3′ and 5′-GGCCTGGTTGAGGGTCACT-3′; CHOP, 5′-TGCCTTTCTCTTCGGACACT-3′ and 5′-GTCCTCATACCAGGCTTCCA-3′; GAPDH, 5′-GGAGTCAACGGATTTGGTCGTA-3′ and 5′-GCAACAATATCCACTTTACCAGAGTTAA-3′.

### 2.7. Statistical Analysis

Statistical analysis was performed by an unpaired Student’s *t*-test. The *p*-values were set at < 0.01.

## 3. Results

### 3.1. dC17/C18:0 and tC17/C18:0 Cer Preparation

In order to distinguish exogenous Cer from endogenous Cer, we synthesized both N-stearoyl-D-erythro-dihydrosphingosine (C17 base) (for 17NDS synthesis) and N-stearoyl-D-erythro-4-hydroxydihydrosphingosine (C17 base) (for 17NP synthesis), which are undetectable in human skin. LC-MS/MS analysis confirmed that 17NDS and 17NP were successfully synthesized (purity > 99%) (Appendix A).

### 3.2. Conversion of 17NDS and 17NP to Other Cer Species

Heterogenous Cer species are synthesized in the late stages of differentiated KCs. An organotypic reconstituted epidermal model (3D KC cultures) can recapture this heterogenous Cer production. However, we previously established a submerged cultured KC model, which recaptures the epidermis (a quasi-cultured epidermal model), i.e., KC consisting of proliferating and early and late stages of differentiated KC (which are stratified [piled-up]) [30,31,35]. Heterogenous Cer are then produced (a quasi-epidermal model) [30,31,35]. We confirmed the formation of lamellar membrane structures and corneocyte lipid envelopes, both present in the stratum corneum [31]. The advantage of this cultured system is that large amounts of cells are relatively easy to obtain and reproduce. In addition, because insertion into a membrane, used for the conventional 3D KC model, is not used here, cells can be observed under microscopy. We employed this established KC culture system [30,31,35] in this current study. KCs started to incubate with 17NDS and 17NP two days after culturing in the late-stage differentiation induction medium (Appendix A). Increases in K10 keratin (early KC differentiation marker) [36] expression and involucrin (middle KC differentiation marker) and loricrin (late KC differentiation marker) production [30] became evident in cells following the switch to a differentiation induction medium. These results confirm that KC used in these studies is fully differentiated and is capable of synthesizing heterogenous Cer species.

#### 3.2.1. N-Deacylation of 17NDS and 17NP

17dihydrosphingosine (17DS), 17sphingosine (17S), and 4-hydroxy 17dihydrosphingosine (17Po) were generated in KC incubated with 17NDS (Figure 1A), while only 17P was present in KC following 17NP treatment (Figure 2A). In addition, 17DS-1-phosphate (17DS1P) and 17P-1-phosphate (17P1P) are produced from KC treated with 17NDS and 17NP, respectively (Figure 2A). These results suggest that both exogenous 17NDS and 17NP are deacylated to produce sphingoid bases by ceramidase, and some of the sphingoid base derived from 17NDS and 17NP is phosphorylated by sphingosine kinase. Moreover, since P is detected in KC incubated with NDS, generated DS from NDS is used for NDS synthesis, and then metabolized to NS and NP by desaturase/sphingoid ∆ (4) desaturase 1 (DEGS1) and sphingolipid C-4 hydroxylase/sphingolipid ∆ (4)-desaturase 2 (DEGS2), respectively (Figure 1).

#### 3.2.2. Production of 17NS and 17NP

17NDS containing C16, C18, C24, and C24:1 as amide-linked FA were detected in KC incubated with 17NDS (Figure 2B). In addition, 17NS and 17NP containing C16, C18, C24, and C24:1 were produced in KC incubated with 17NDS (Figure 2C,D). 17NP, but neither 17NDS nor 17NS containing C16, C18, C24, and C24:1 as amide-linked FA, were detected in KC incubated with 17NP (Figure 2B–D). These results suggest that 17NDS and 17NP are hydrolyzed to DS and P, and FA, respectively. The generated 17sphingoid base and pooled FA are then used for Cer synthesis (Figure 1). Moreover, resynthesized 17NDS is converted to 17NS and 17NP. However, 17P derived from 17NP is only used for 17NP synthesis.

#### 3.2.3. Production of Omega-O-AcylCers

Both EOS (C30, C32, and C32:1) and EOP (C30, C32, and C32:1) were produced in KC incubated with 17NDS, while only 17EOP (C30, C32, and C32:1) were generated in 17NP-treated cells (Figure 2E,F).

Note that, following molecular remodeling, i.e., conversion to various sphingoid bases and amide-linked fatty acids, certain amounts of Cer are converted to glycosylated Cer (mainly glucosylCer), omega-O-acylglucosylCers, and sphingomyelin.

### 3.3. Alterations of Cer Levels in KC Incubated with NDS and NP

Cer production is significantly increased in the late stage of differentiated KCs [31,37,38]. Since it is uncertain whether exogenous Cer contributes to further increases in epidermal Cer levels, we next investigated this in KCs using N-stearoyl-D-erythro-dihydrosphingosine (18NDS) and N-stearoyl-D-erythro-4-hydroxydihydrosphingosine (18NP), which are two of the endogenous Cer species in KC.

Total Cer (NDS, NP, and NS) was significantly increased in KC treated with both NDS and NP (250 µM), while statistically significant increases in NS, NP, and NS were observed in KC incubated with 50 µM of NDS (Figure 3). Only NP was significantly increased in KC treated with incubated 50 µM NP.

Moreover, both EOS and EOP are significantly increased in cells treated with NDS, while NP increased EOP, but not EOS (Figure 3D,E). S, ND, and P were modestly increased in KC treated with 18NS and 18NP, respectively (Figure 3F). These results suggest that exogenous Cer (>50 µM) is capable of further elevating Cer and its metabolite levels in the late stage of differentiated KCs. Thus, the supplementation of exogenous Cer should be effective in increasing epidermal Cer levels.

### 3.4. Modulation of KC Differentiation by Exogenous Cer

We next investigated whether exogenous NDS and NP modulate KC differentiation. Many protein levels, including structural proteins and enzymes, are changed during KC differentiation. Since the aim of our study is to describe the effect of exogenous Cer on KC differentiation, we measured established KC differentiation markers’ protein, i.e., keratin 10, involucrin, and loricrin in early, mid, and late stages of KC differentiation, respectively. Both keratin 10 and involucrin protein levels are increased in vehicle-treated KC on day 3, while loricrin levels are increased on day 5 (Figure 4A). Increases in loricrin levels in cells treated with both NDS and NP were higher than the control, while involucrin levels were higher in NDS-treated cells at 3 and 5 days vs. NP and control (Day 3: 1.9-fold and 1.4-fold, vs. vehicle and NP, respectively) and Day 5 (2.8-fold and 1.3-fold, respectively) (Figure 4A).

These results suggest that both exogenous NDS and NP, and/or their metabolite(s) can promote KC differentiation. In addition, NDS and/or its metabolites likely specifically stimulate involucrin production independent of global stimulation of differentiation. 

### 3.5. Modulation of Cathelicidin Antimicrobial Peptide by Exogenous Cer

Cathelicidin antimicrobial peptide (CAMP/LL-37) is a multi-functional peptide [39,40,41] that not only includes antimicrobial activity, but also modulates cell motility and differentiation [39,40,41]. Sufficient production of cathelicidin occurs in atopic dermatitis in response to pathogenic microbial infection [39,40,41]. We previously found that S1P, but not DS1P, stimulates cathelicidin antimicrobial peptide (CAMP/LL-37) [30,32,33]. Sphingoid base-1-phosphate was increased in KC incubated with NDS and NP (Figure 2A and Figure 3A). Hence, we investigated whether S1P generated from exogenous Cer suffices to promote this CAMP synthesis. As shown in our prior studies [42], CAMP production is increased by differentiation, while NDS, but not NP, further elevated CAMP (Figure 4B). These results suggest that increases in S1P by exogenous NDS could promote CAMP production.

## 4. Discussion

Our study using 17NDS and 17NP characterized the metabolic fate of exogenous Cer in the late stages of differentiated KC as helping to recapture features of healthy epidermis, including synthesizing diverse Cer species required for epidermal permeability barrier formation [3,4]. We demonstrated that structural (compositional) remodeling of exogenous 17NDS and 17NP occurs in KC. Exogenous Cer is hydrolyzed to the sphingoid base and FA, and the generated sphingoid base and a pool of FA in cells become precursors to Cer synthesis. Cer species consisting of different N-acyl chain lengths of FA and P and P1P are produced from NP, while, as expected, diverse species of Cer can be produced from NDS, but not NP. Although Cer species containing ß-hydroxy FA were not analyzed, as shown in Figure 1, BS can be produced by the metabolic conversion of N-acyl FA. Yet, since 6-hydroxy sphingosine and 4,14-sphingadiene synthetic enzymes are not completely identified, it remains to be resolved if NH and NSD are generated from NDS. 

Since Cer production is significantly elevated in the late stage of differentiated KCs, it is possible that negligible amounts of Cer are produced from exogenous Cer. We confirmed that exogenous Cer further increases Cer levels, suggesting the pharmacological relevance of exogenous Cer to elevate Cer levels in the late stages of KC. We previously found that S1P (P1P is less potent vs. S1P [unpublished data]) stimulates the synthesis of a key epidermal antimicrobial peptide, CAMP [30,33]. We here showed that exogenous NDS increases CAMP production, suggesting the amounts of S1P from NDS (NDS→NS→S→S1P and NDS→DS→NDS→NS→S→S1P) are enough to signal CAMP production. These results further suggest that the exogenous application of Cer suffices to increase biologically relevant levels of Cer and its metabolites in cells. In particular, NDS has the ability to increase Cer production at a lower concentration (50 µM). Thus, suitably formulated Cer, which enhances the absorption of Cer [29], is useful for increased endogenous Cer production, leading to improving epidermal permeability barrier integrity and antimicrobial barrier defenses.

Cer and its metabolites modulate cellular proliferation and differentiation in KC [24,43,44]. Kim et al. reported that P promotes differentiation in KC through the transactivation of PPAR [45]. Because NP is decreased in atopic dermatitis and psoriasis, P was a focus of their study [45]. Sigruener showed that not only P but also ND are potent sphingoid bases that induce KC differentiation [46]. Taken together with our current study, not only P but also other sphingoid bases are able to increase KC differentiation. We showed here that NDS is more effective in increasing involucrin levels (Figure 4), and NDS and its metabolite(s) may specifically upregulate involucrin. Although it remains to be resolved (1) which molecule (either Cer or its metabolites) it is and (2) how this particular molecule (derived from NDS) promotes involucrin synthesis, our results suggest that specific step(s) or protein(s) associated with differentiation are regulated by Cer and/or its metabolites.

Like NP, NS could be a precursor for Cer-containing S synthesis and stimulate KC differentiation. However, relative amounts of NS are increased in atopic dermatitis. In addition, we previously reported that a ratio of S to DS is increased in the SC of an atopic dermatitis mouse model, and changes in the ratio of S to ND alter the packing of lamellar bilayers, i.e., increases in the ratio of S to DS decrease tightly packed orthorhombic structure [47]. Because vital acidic and alkaline ceramidase are present in the SC [48,49], DS produced from NDS may modulate the ratio of S/ND related to the packing of lamellar bilayers, and may result in increasing the integrity of the lamellar membrane in the SC [47]. Conversely, S generated from NS may affect orthorhombic structures in SC to attenuate barrier integrity.

The excessive production of Cer induces apoptosis in cells, including in KCs [43], but we did not see any toxic effects using exogenous NDS and NP (up to 250 µM) in our KC studies. In addition, apoptotic activity through Cer channel formation of NS is stronger than with NDS [50] (note that P was not tested). Thus, NDS might be less toxic in cells. However, since NS is the Cer species used for most studies that investigate apoptotic effects of Cer in cells/tissues, the deleterious effects of NDS have not been completely elucidated [51]. Hence, optimal amounts of any Cer in formulations should be carefully considered to maximize positive outcomes in vivo.

Insights gained from our study propose that topical Cer agents are classified into two groups. Group 1 includes Cer agents that improve the epidermal barrier on the skin surface and in the SC. Group 2 includes Cer agents that work in KCs in nucleated layers, i.e., endogenous Cer synthesis increases via increases in Cer synthesis precursors and by promoting KC differentiation. Cer agents used for barrier-compromised skin and formulated with skin absorption enhancers are categorized in Group 2. Groups 1 and 2 can also be classified as first and second generations of topical Cer agents, respectively (Figure 5).

Our study could become a basis for developing topical agents containing Cer. Since we now know the fate of exogenous Cer and the effect of exogenous Cer on KC differentiation, we could test the effects of exogenous Cer in an appropriate formulation (that enhances the penetration of Cer into the stratum corneum) on epidermal barrier function. Using in vivo/ex vivo human skin.

## 5. Conclusions

Increased epidermal Cer is a therapeutic strategy to improve epidermal permeability barrier function. We demonstrated that exogenous NDS has a greater ability as a precursor to diverse Cer species required for the vital permeability barrier production than exogenous NP has. Moreover, although both NDS and NP and their metabolites enhance KC differentiation and increase endogenous Cer, NDS increases S1P levels, leading to upregulated cathelicidin (innate immune component) production. Because NDS is a minor component of the SC, this Cer species has not previously received a great deal of attention. Our current study demonstrates that NDS could be a multi-potent Cer species for epidermal barrier formation and a lipid mediator that stimulates KC differentiation and enhances innate immunity (Appendix A).

## Figures and Tables

**Figure 1 cells-11-01742-f001:**
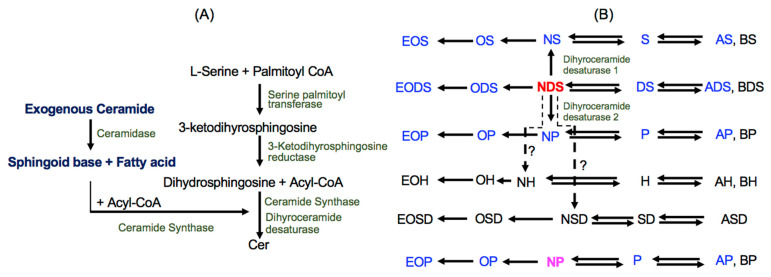
**Cer synthetic pathway and fates of exogenous NDS and NP.** (**A**) Ceramide synthetic pathway, (**B**) Fates of NDS and NP. Abbreviations for nomenclature of Cer species are according to [2,11,25] with some modifications. Non-hydroxy acyl sphingosine (NS); Non-hydroxy acyl 4-hydroxydihydrosphingosine (NP); Non-hydroxy acyl dihydrosphingosine (NDS); Non-hydroxy acyl 6-hydroxydsphingosine (NH); Non-hydroxy acyl 4,14-sphingadiene (NSD); 2-hydroxy acyl sphingosine (AS); 2-hydroxy acyl 4-hydroxydihydrosphingosine (AP); 2-hydroxy acyl dihydrosphingosine (ADS); 2-hydroxy acyl 6-hydroxy sphingosine (AH); 2-hydroxy acyl 4,14-sphingadiene (ASD); ß-hydroxy acyl sphingosine (BS); ω-hydroxy-N-non-hydroxy acyl sphingosine (OS); ω-hydroxy-N-non-hydroxy acyl 4-hydroxydihydrosphingosine (OP); ω-hydroxy-N-non-hydroxy acyl dihydrosphingosine (ODS), ω-hydroxy-N-non-hydroxy acyl 6-hydroxydsphingosine (OH); ω-hydroxy-N-non-hydroxy acyl 4,14-sphingadiene (OSD); ω-O-acyl-N-non-hydroxy acyl sphingosine (EOS); ω-O-acyl-N-non-hydroxy acyl 4-hydroxydihydrosphingosine (EOP); ω-O-acyl-N-non-hydroxy acyl dihydrosphingosine (EODS), ω-O-acyl-N-non-hydroxy acyl 6-hydroxydsphingosine (EOH); ω-O-acyl-N-non-hydroxy acyl 4,14-sphingadiene (EOSD). Cer species confirmed in current study are in blue. Cer in black letters is not analyzed.

**Figure 2 cells-11-01742-f002:**
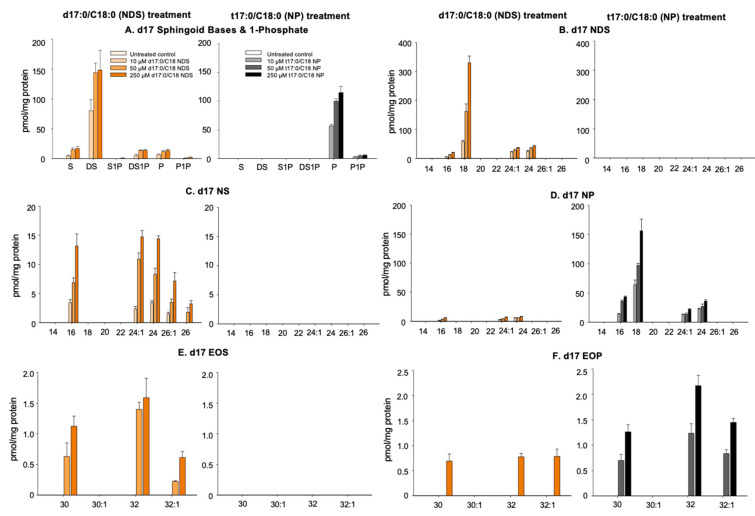
**Fate of exogenous 17Cer and 17NP.** Cer species containing ß-hydroxy FA, 6-hydroxy sphingosine (H), 4,14-sphingadiene (SD), and omega-hydroxy Cer were not analyzed. *n* = 3, * *p* < 0.01 vs. Mean ± SD. see details in Section 2.

**Figure 3 cells-11-01742-f003:**
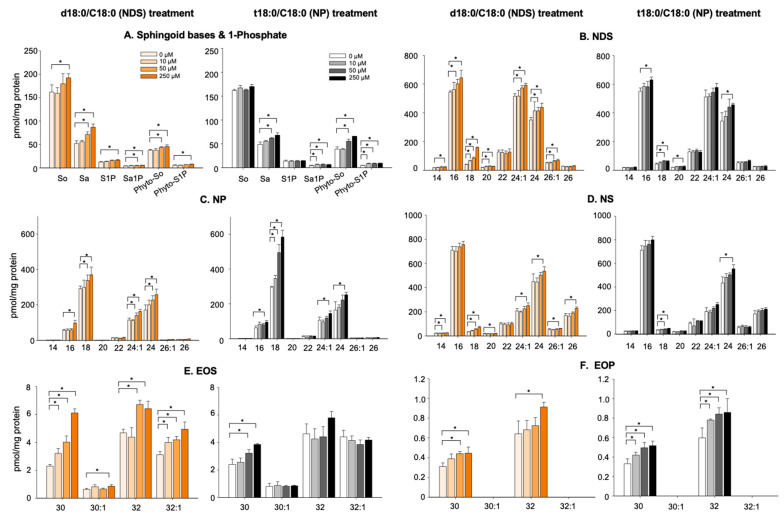
**Exogenous Cer further increases Cer levels in late stage of differentiated KC**. Cer species containing ß-hydroxy FA, 6-hydroxy sphingosine (H), 4,14-sphingadiene (SD), and omega-hydroxy Cer were not analyzed. *n* = 3, Mean ± SD. * *p* < 0.01 *vs* indicated group. see details in Materials and Methods.

**Figure 4 cells-11-01742-f004:**
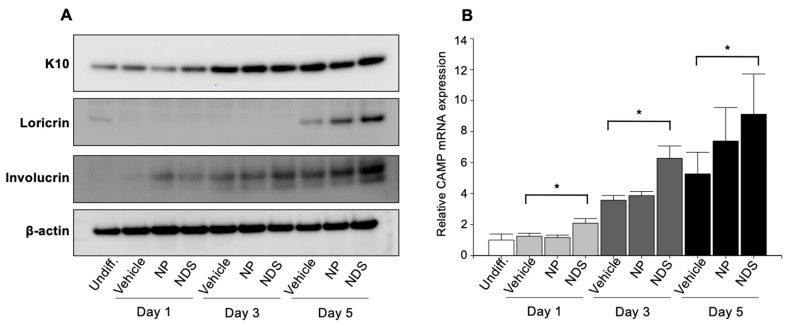
**Exogenous ceramide stimulates KC differentiation and increases CAMP mRNA expression.** (**A**) Western blot analysis (cultured in 0.07 mM Ca^2+^ serum-free KC medium). (**B**) CAMP mRNA levels were assessed by *q*RT-PCR. Cells are incubated with 250 µM of NP or NDS. Undiff = undifferentiated KC. *n* = 3, Mean ± SD, * *p* < 0.01 vs. vehicle. Please see details in Material and Methods.

**Figure 5 cells-11-01742-f005:**
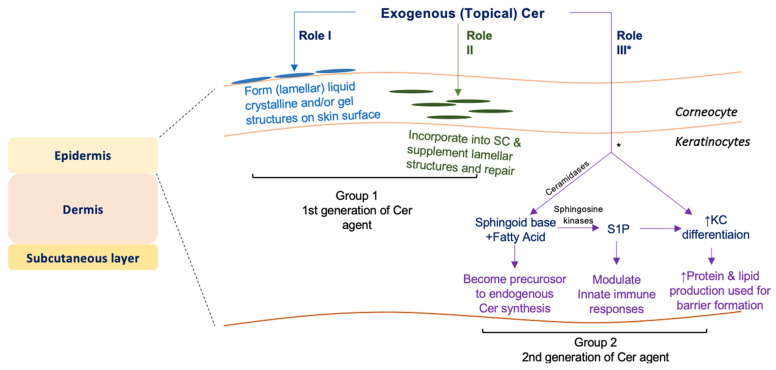
Classification of topical Cer agents.

## Data Availability

Data are contained within the article and Appendix A.

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
