# Peer review of "Exogenous Ceramide Serves as a Precursor to Endogenous Ceramide Synthesis and as a Modulator of Keratinocyte Differentiation"

_cells, 2022, doi:10.3390/cells11111742_

Round 1
Reviewer 1 Report
In this manuscript, Shin and co-authors report on ceramide content and protein levels of differentiation markers in cultured human primary keratinocytes (KC) which were supplemented with exogenous ceramide (Cer) species during differentiation process. They synthetized exogenous CerNDS and CerNP containing 17C-sphingoid bases, which are not physiological to human cells and are sometimes used as internal standards for analytical methods. Supplementation with exogenous 17Cer in differentiated KC increased their endogenous Cer levels. Authors also show that exogenous 17Cer (or at least its part) is hydrolysed in KC and released sphingoid bases are used in salvage pathway of Cer synthesis.
General concept comments:
There is not clearly stated what is the aim of the study, several times a topical application of ceramides or skin care products containing them is mentioned, however the relevance to presented experiments is not well-explained. Statements are sometimes contradictory and without proper citations (e.g. line 91-93). - Can you comment on the Cer composition (sphingoid base and chain length of amide-linked Fa) of the already used topical Cer and why have you decided to use Cer 17:0/18:0?
Cathelicidin is firstly mentioned in the Material and methods and then in short paragraph in Results section. What is the role of cathelicidin in skin diseases you mention? Add info to the introduction and discuss later properly. Is the increase of this antimicrobial peptide always beneficial in skin pathologies?
KC model system: I would appreciate better description of the model (could be in Supplementary info) as it differs from the “conventional3D KC model” (line 197). Is the here used model considered to be 3D or 2D model? You state that it is stratified - how many layers of KC does it have? Does it contain corneocytes in the uppermost layer of KC if they are submerged in medium? Can you provide some illustrative pictures, how the model looks like? How do you harvest the cells?
I appriaciate the synthesis of exogenous shinogid bases and subsequently Cer they performed. It enables to distinguish them from endogenous Cer species and with the presented methodology they showed that the synthetized Cer are recognized by KC as substrates for enzymatic Cer remodeling. Can you calculate and discuss how much of the exogenous 17Cer are hydrolyzed and transformed to other 17Cer species?
Levels of selected endogenous Cer classes were evaluated. Discuss the individual species composition, meaning the acyl chain length and how they change depending on the dose of exogenous 17Cer. Do you have a mechanistic explanation, why the exogenous Cer increase the endogenous Cer levels? Do they increase d18Cer de novo synthesis or promote d18Cer release from their precursors? The article could profit from an additional functional experiment.
No statements on topical Cer application should be done based on this study. It is described that the exogenous Cer can increase Cer levels at level of living differentiating KC in culture, but no topical application neither penetration through or any influence on permeability barrier was tested here, e.g. line 353-355: “… exogenous NDS has a greater ability as a precursor to diverse Cer species for a vital permeability barrier production …” Do you mean by exogenous NDS the 17NDS? It is still questionable, how 17Cers substitute endogenous Cer in the barrier formation and its function.
Specific comments:
Introduction, line 43-44: please reformulate the sentence: “The SC consists of heterogeneous Cer molecular species, ...” – e.g. “Cer fraction of SC extracellular matrix” or similar.
Introduction, line 53: please be careful with using words like “treat” in connection to the topical products which only mitigate the disease consequences/symptoms.
Shortening of sphingoid bases as So, Sa and PhytoSo is rather uncommon. I recommend to use the deep-rooted labeling as it is used in nomenclature of Cer here – sphingosine = S, sphinganine = DS, phytosphingosine = P.
Figure 1, line 85: typo – there is only OH instead of EOH
Figure 1: why are there displayed beta-hydroxylated Cer? There is no comment on them in introduction, only later mentioned that they were not evaluated. So is there any relevance/importance to the presented paper and/or impaired skin barrier? If yes, included it in introduction, if not exclude them from the Figure 1.
Material and Methods, 2.2 Cell culture: please add the final concentration of exogenous Cer in culture media also here and comment on the dilution factor for ethanol.
Material and Methods, 2.4 Lipid analysis: add short information of the method used for protein quantification.
Results, paragraph 3.2: how is it relevant to this study, that the differentiated KC model can be observed under microscopy?
3.2.2, line 223-224: “These results suggest that Sa and PhytoSo derived from 17NDS and 17NP are hydrolyzed to 17Sa and 17PhytoSo, respectively.” Reformulate please.
3.3, line 237: provide more citations on Cer production in late stage of differentiated KC
Please unify the labeling of exogenous Cer as 17NDS and 17NP, especially in Results, paragraph 3.3 it is confusing (e.g. line 242-245: “Total Cer (NDS, NP and NS) was significantly increased in KC treated with both NDS and NP (250 µM), …”). Consider labeling the endogenous Cer everywhere as 18NDS etc.
Discussion, line 326: “precursors for Cer-containing So synthesis” ?? – you probably mean So-containing Cer synthesis, or?
Line 350 – Figure 5 is missing
Author Response
Comment 1: There is not a clearly stated aim of the study, and several times a topical application of ceramides or skin care products containing them is mentioned, however the relevance to presented experiments is not well-explained. Statements are sometimes contradictory and without proper citations (e.g. line 91-93). –
Response: We addressed the aim of our study in the last part of the Introduction.
“Here, we aim to define the fates of exogenous Cer, the contribution of exogenous Cer to increased Cer levels in differentiated KC, and the effect of exogenous Cer on KC differentiation.”
In addition, we revised “Stimulation of endogenous Cer production is already a therapeutic strategy used to improve epidermal permeability barrier integrity.”
Comment 2: “Can you comment on the Cer composition (sphingoid base and chain length of amide-linked Fa) of the already used topical Cer and why have you decided to use Cer 17:0/18:0?”
Response: Cer with a C17 sphingoid base are not used as ingredients in commercial skin care products. Cer with a C18 sphingoid base are used as ingredients, including:
NP: N-stearoyl phytosphingosine and N-Oleoyl phytosphingosine),
AP N-2-hydroxystearoyl phytosphingosine
NDS: N-stearoyl dihydrosphingosine
ADS: N-palmitoyl dihydrosphingosine
EOP: N-(27-Octadecanoyloxy-heptacosanoyl) phytosphingosine
As described in “Section 3.1. dC17/C18:0 and tC17/C18:0 Cer preparation”, we used C17Cer containing a C17 sphingoid base to distinguish from endogenous Cer.
We have revised the first sentence of 3.1. dC17/C18:0 and tC17/C18:0 to clarify the purpose for using 17:0/18:0Cer.
“In order to distinguish exogenous Cer from endogenous Cer, we synthesized both N-stearoyl-D-erythro-dihydrosphingosine (C17 base) (for 17NDS synthesis) and N-stearoyl-D-erythro-4-hydroxydihydrosphingosine (C17 base) (for 17NP synthesis), which are undetectable in human skin.”
We clarify that Cer 18:0 (dihydrosphingosine[sphinoganine])/18:0 (NDS) and C18:0 (4- dihydrosphingosine [phytosphingosine]) are used in studies in Sections 3.3, 3.4 and 3.5.
Comment 3: Cathelicidin is firstly mentioned in Materials and Methods and then in a short paragraph in Results section. What is the role of cathelicidin in skin diseases you mention? Add info to the introduction and discuss later properly. Is the increase of this antimicrobial peptide always beneficial in skin pathologies?
Response: As stated in section 3.5. Modulation of cathelicidin antimicrobial peptide by exogenous Cer, the purpose of assessing CAMP/LL-37 is:
“Although several antimicrobial peptides are produced in KC, we previously found that S1P, but not Sa1P, stimulates cathelicidin antimicrobial peptide (CAMP/LL-37) [1-3]. Sphingoid base-1-phosphate was increased in KC incubated with NDS and NP (Figs. 2A and 3A). Hence, we investigated whether S1P generated from exogenous Cer suffices to promote this CAMP synthesis.” Following this reviewer’s comment, we include a brief feature of CAMP/LL-37:
“Cathelicidin antimicrobial peptide (CAMP/LL-37) is a multi-functional peptide [4-6] that not only antimicrobial activity, but also modulate cell motility and differentiation [4-6]. Sufficient production of cathelicidin occurs in atopic dermatitis in response to pathogenic microbial infection [40-42].
Comment 4: KC model system:
1) I would appreciate a better description of the model (could be in Supplementary info) as it differs from the “conventional 3D KC model” (line 197). Is the model used here considered to be 3D or 2D model?
Response: Our cells are not plated on membrane, collagen, or fibroblasts. Thus, this model is different from conventional 3D KC. Our model KCs are piled up, resulting in top layer cells are localized at the interface between air and liquid. Since our model can be considered as 3D, we described it as a pseudo-epidermal model.
2) You state that it is stratified - how many layers of KC does it have?
Response: We have attached our prior electron microscopy photographs (Uchida Y., et al. J Invest Dermatol 2001).
Cross section of our model keratinocytes
Arrows indicate lamellar bodies
This photo image shows at least three layers.
In addition, cornified envelope bound lipid, omega-OH glucosylceramide, omega-OH ceramide, and omega-OH fatty acids are present in our model KC (VitC in Table II, below). Cornified envelope bound lipid formation: First omega-OH glucosylceramide is bound to cornified envelope, and then beta-glucosidase cleaves glucose to form omega-OH ceramide, and finally ceramidase hydrolyzes ceramide. Therefore, enzymatic lipid processing occurring during transition of stratum granulosum and in the stratum corneum is recaptured in our model.
N-omega-OH-Cer-A: (EOS) and N-omega-OH-Cer-B: EOP
(Uchida Y., et al. J Invest Dermatol 2001).
4) Does it contain corneocytes in the uppermost layer of KC if they are submerged in medium?
Response: As above, Yes.
5) Can you provide some illustrative pictures, what the model looks like?
Response: Since we have already published microscopic images, we refer to these prior studies in our current manuscript.
We published the following picture (Shin KO et al., J Invest Dermatol 2017). UDK: undifferentiated, EDK: early stage of differentiation and LDK: late stage of differentiation. LDK, some cells are rectangular shaped (stratified cell shape)
6) How do you harvest the cells?
Response: Cells are harvested using cell scrapers. Similar to epidermis, cells are recovered as sheets.
Comment 5: I appreciate the synthesis of exogenous shingoid bases and subsequent Cer. It distinguishes them from endogenous Cer species and with the presented methodology they showed that the synthetized Cer are recognized by KC as substrates for enzymatic Cer remodeling.
1) Can you calculate and discuss how much of the exogenous 17Cer are hydrolyzed and transformed to other 17Cer species? Levels of selected endogenous Cer classes were evaluated. Discuss the individual species composition, meaning the acyl chain length and how they change depending on the dose of exogenous 17Cer.
Response:
We clarify that the aim of our study is to characterize the different fates of NDS and NP, focusing on molecular remodeling of different Cer species (sphingoid base and amide-linked fatty acid). We calculated hydrolyzed amounts of exogenous Cer and remodeling Cer amounts.
But generated Cer is converted to not only Cer, but also glycosylated Cer and sphingomyelin.
Calculated data do not represent how much of the exogenous 17Cer are transformed to other 17Cer species. Hence, we do not report hydrolysis and remodeling rates of exogenous Cer in our manuscript.
In our revision, we include: “Note: Following molecular remodeling; i.e., conversion to various sphingoid bases and amide-linked fatty acids, certain amounts of Cer are converted to glycosylated Cer (mainly glucosylCer) and sphingomyelin” in the last part of Section 3.2.2.
10 uM exogenous 7NDS and 17NP
84% of exogenous 17NDS and 62% of 17NP are hydrolyzed.
20% of exogenous 17NDS and 29% of 17NP are converted to different Cer species (fatty acid chain lengths after hydrolysis).
50 uM exogenous 7NDS and 17NP
65% of exogenous 17NDS and 65% of 17NP are hydrolyzed.
25% of exogenous 17NDS and 29% of 17NP are converted to different Cer species (fatty acid chain lengths after hydrolysis).
250 uM exogenous 7NDS and 17NP
53% of exogenous 17NDS and 51% of 17NP are hydrolyzed.
24% of exogenous 17NDS and 15% of 17NP are converted to different Cer species (fatty acid chain lengths after hydrolysis).
A dose-dependent trend that decreases hydrolytic rate may be due to decreases in uptake of exogenous Cer in cells, but this has not been proven.
2) Do you have a mechanistic explanation, why the exogenous Cer increase the endogenous Cer levels? Do they increase d18Cer de novo synthesis or promote d18Cer release from their precursors? The article could profit from an additional functional experiment.
Response: Exogenous Cer can be utilized as a substrate of Cer production in keratinocytes after hydrolysis. Since increases in precursors increase Cer production, exogenous Cer could contribute to increased precursors. Moreover, Cer production is increased during keratinocyte differentiation as described in our manuscript. Thus, exogeneous Cer stimulates differentiation, thereby increasing total Cer levels.
Comment 6: No statements on topical Cer application should be made based on this study. It is described that the exogenous Cer can increase Cer levels at level of living differentiating KC in culture, but no topical application or penetration through or any influence on permeability barrier was tested here, e.g. line 353-355: “... exogenous NDS has a greater ability as a precursor to diverse Cer species for a vital permeability barrier production ...” Do you mean by exogenous NDS, the 17NDS? It is still questionable, how 17Cer substitutes for endogenous Cer in the barrier formation and its function.
Response: As described in our Introduction, a formula to enhance efficacy of exogenous Cer has being developed, suggesting certain amounts of topically applied Cer can penetrate into skin. However, the fate of exogenous Cer has not been completely defined. The aim of our current study is to elucidate the fate of exogenous Cer and also the ability of exogenous Cer to modulate keratinocyte differentiation. Following this reviewer’s comment, we clarify the aim of our study in the Introduction. Our study could become a basis for developing topical agents containing Cer. To assess whether topical Cer contributes to increases in Cer levels in the stratum corneum and keratinocyte differentiation using in vivo/ex vivo human skin is not the aim of our current study. Since we now know the fate of exogenous Cer and the effect of exogenous Cer on KC differentiation, we could test the effects of exogenous Cer in an appropriate formulation (that enhances the penetration of Cer into the stratum corneum) on epidermal barrier function.
Specific comments:
Comment 7: Introduction, line 43-44: please reformulate the sentence: “The SC consists of heterogeneous Cer molecular species, ...” – e.g. “Cer fraction of SC extracellular matrix” or similar.
Response: Following the reviewer’s comment, we revised this sentence.
The Cer in the SC consisting SC consists of heterogenous Cer molecular species (21 molecular groups are identified in humans) [7].
Comment 8: Introduction, line 53: please be careful with using words like “treat” in connection to the topical products which only mitigate the disease consequences/symptoms.
Response: Thank you - we revised this sentence.
Since topical application of Cer improves epidermal permeability function [8-10], Cer is formulated as an active ingredient for topical products to treat attenuate disease symptoms skin diseases associated with compromised permeability barrier function.
Comment 9: Shortening of sphingoid bases as So, Sa and PhytoSo is rather uncommon. I recommend using the deep-rooted labeling as it is used in nomenclature of Cer here – sphingosine = S, sphinganine = DS, phytosphingosine = P.
Response: Following the reviewer’s suggestion, we have used their abbreviations in our revision.
Comment 10: Figure 1, line 85: typo – there is only OH instead of EOH
Response: Authors checked this line. However, OH indicates ω-hydroxy-N-non-hydroxy acyl 6-hydroxydsphingosine. OH is not a typo.
Comment 11: Figure 1: why are beta-hydroxylated Cer displayed? There is no comment on them in the introduction, only later mentioned that they were not evaluated. So is there any relevance/importance to the presented paper and/or impaired skin barrier? If yes, include it in introduction, if not exclude them from the Figure 1.
Response: The presence of beta-hydroxylated Cer was reported in 2020 by Kawana M. et al (Ref #2). This physiological and pathological role of beta-hydroxylated in skin are unknown. Although we did not analyze beta-hydroxylated Cer, we believe including this Cer species is useful in understanding an overview of Cer in the SC for readers.
Comment 12: Materials and Methods, 2.2 Cell culture: please add the final concentration of exogenous Cer in culture media here and comment on the dilution factor for ethanol.
Response: Information on final concentrations of Cer are reported in Figure legends.
Comment 13: Materials and Methods, 2.4, Lipid analysis: add information of the method used for protein quantification.
Response: We have included information on our protein assay in Materials and Methods, 2.4.
Comment 14: Results, paragraph 3.2: how is it relevant to this study, that the differentiated KC model can be observed under microscopy?
Response: Please see response to Comment 4, above.
Comment 15: 3.2.2, line 223-224: “These results suggest that Sa and PhytoSo derived from 17NDS and 17NP are hydrolyzed to 17Sa and 17PhytoSo, respectively.” Reformulate please.
Response: We have revised this sentence.
These results suggest that 17NDS and 17NP are hydrolyzed to DS and P, and fatty acid, respectively.
Comment 16: 3.3, line 237: provide more citations on Cer production in late stage of differentiated KC.
Response: We have cited relevant original prior articles.
Comment 17: Please unify the labeling of exogenous Cer as 17NDS and 17NP, especially in Results, paragraph 3.3 is confusing (e.g., line 242-245: “Total Cer (NDS, NP and NS) was significantly increased in KC treated with both NDS and NP (250 µM), …”). Consider labeling the endogenous Cer everywhere as 18NDS etc.
Response: As above, 17NDS and 17NP are used in studies discussed in Section 3.2. In Section 3.3, we stated “Cer production is significantly increased in the late stage of differentiated KCs [11-13]. Since it is uncertain whether exogenous Cer contributes to further increases in epidermal Cer levels, we next investigated this in KCs using N-stearoyl-D-erythro-dihydrosphingosine (18NDS) and N-stearoyl-D-erythro-4-hydroxydihydrosphingosine (18NP), which are two of the endogenous Cer species in KC.” We believe that further labeling is unnecessary in the text.
Comment 18: Discussion, line 326: “precursors for Cer-containing So synthesis” ?? – you probably mean So-containing Cer synthesis, or?
Response: We reworded Cer-containing S synthesis
Comment 19: Line 350 – Figure 5 is missing
Response: We apologize for the missing figure. We have now added Figure 5 to our revision.

Reviewer 2 Report
- Page 3. Line 123. The authors purified 17NDS or 17NP by using silica gel column chromatography. Please provide the brand and pore size (specification) of the silica gel.
- Page 5. Line 185. The 17NDS and 17NP were confirmed by LC-MS/MS. The authors showed the MS profile in Supp. Fig 2. Please add the HPLC chromatogram and provide the purity of the final products.
- Page 8. Figure 4. Please explain Fig 4(A) Western blot analysis, why the is Involucrin shown in double bends?
- Page 9. Line 350. We cannot find Figure 5 in the manuscript.
Author Response
Reviewer 2
Comment 1: Page 3. Line 123. The authors purified 17NDS or 17NP by using silica gel column chromatography. Please provide the brand and pore size (specification) of the silica gel.
Response: We have included information of the silica gel in revision.
Silica gel for column chromatography, particle size: 0.040-0.063 mm, pore size: 60 Å (Sigma-Aldrich, St. Louis, MO.)
Comment 2: Page 5. Line 185. The 17NDS and 17NP were confirmed by LC-MS/MS. The authors showed the MS profile in Supp. Fig 2. Please add the HPLC chromatogram and provide the purity of the final products.
Response: We have included HPLC chromatogram and the purity of the final products in revision. One peak was detected on HPLC chromatogram of both 17NDS and 17NP (Supplemental Figure 3). Purity of 17NDS and 17NP is over 99%.
Comment 3: Page 8. Figure 4. Please explain Fig 4(A) Western blot analysis, why the is Involucrin shown in double bends?
Response: We believe second band (lower molecular weight band) is non-specific band.
Comment 4: Page 9. Line 350. We cannot find Figure 5 in the manuscript.
Response: We apologize for the missing figure. We have now added Figure 5 to our revision.
Reviewer 3 Report
Shin et al. examined the effect of acyl ceramide species on keratinocyte differentiation, a coordinated process of lipid metabolism and cytoskeletal cross-linkages utilizing effectors such as involucrin/loricrin in vitro.
The strength of this study is that they tested non-hydroxy acyldihydrosphingosine (NDS) and non-hydroxy acyl 4-hydroxydihydrosphingosine (NP), commonly used ingredients in skincare products. Although NDS is a minor stratum corner component, they have shown its stronger effect on keratinocyte terminal differentiation, involving tissue-intrinsic antimicrobial defense via cathelicidin antimicrobial peptide production.
The manuscript is well-written, and the data and conclusions look valid and may meet the quality that warrants the publication in the journal. This reviewer raises a minor technical issue.
1. Keratinocyte differentiation produces insoluble cell envelopes (CE) predominately composed of involucrin, and the author found the difference in the patterns of CE precursors, which makes sense. The reviewer thinks it is crucial to describe the sample preparation method in the Western blot results (the information in reference [33] was not enough); what kind of buffer (SDS-based or urea/reduced or non-reduced), etc.
Author Response
Reviewer 3
Comment: The reviewer thinks it is crucial to describe the sample preparation method in the Western blot results (the information in reference [33] was not enough); what kind of buffer (SDS-based or urea/reduced or non-reduced), etc.
Response: Agents, gels, and apparatus for Western blot analysis are purchased from Invitrogen. Cell lysates were prepared using radioimmunoprecipitation assay buffer (25 mM Tris-HCl pH 7.6, 150 mM NaCl, 1% NP-40, 1% sodium deoxycholate, 0.1% SDS). All procedures are conducted following manufacturer’s instructions.
We have revised method of Western blot analysis.
Western blot analysis was performed, as described previously [33]. Briefly, cell lysates, prepared in radioimmunoprecipitation assay buffer (RIPA Lysis and Extraction Buffer) were resolved by electrophoresis on 4-12% Bis-Tris protein gel (Invitrogen) under denaturing condition using SDS. All procedures are conducted following manufacturer’s instructions. Resultant bands were blotted onto polyvinylidene difluoride membranes, probed with anti-human keratin 10 (Santa Cruz Biotechnology, Dallas, TX), anti-human involucrin Abcam (Cambridge, UK), or anti-human β-actin (Sigma-Aldrich, St. Louis, MO), and detected using enhanced chemiluminescence (Thermo Fisher Scientific, Waltham, MA). The intensity of bands was measured with a LAS-3000 (Fujifilm, Tokyo, Japan).
Round 2
Reviewer 1 Report
Thank you for addressing my comments. I appreciate the changes done on the manuscript which have improved it. There are only minor changes left that have to be done.
Comment1: Figure 1(B): abbreviations in the figure do not fit the text now (Sa, So, PhytoSo etc.)
Comment 2: from cover letter of authors: ““In our revision, we include: “Note: Following molecular remodeling; i.e., conversion to various sphingoid bases and amide-linked fatty acids, certain amounts of Cer are converted to glycosylated Cer (mainly glucosylCer) and sphingomyelin” in the last part of Section 3.2.2.”” – cannot find it in the uploaded version 2
Comment 3: I appreciate the explanation authors provided to my comments, however some part of these explanations should be still included in the manuscript.
E.g. from the cover letter (response to original comment 6): - “Our study could become a basis for developing topical agents containing Cer. To assess whether topical Cer contributes to increases in Cer levels in the stratum corneum and keratinocyte differentiation using in vivo/ex vivo human skin is not the aim of our current study. Since we now know the fate of exogenous Cer and the effect of exogenous Cer on KC differentiation, we could test the effects of exogenous Cer in an appropriate formulation (that enhances the penetration of Cer into the stratum corneum) on epidermal barrier function.” –
Include its shortened version to the Conclusion.
Comment 4: Discussion, line 354-360: “Our study suggests that topical Cer agents are classified into two groups. …” – do you suggest this classification for the first time or is it already established? Rephrase the sentence to make it clear. (e.g. Based on our study we propose following classification of topical Cer agents…)
Comment 5: I would recommend to check the manuscript for typos and completeness of renaming of the sphingoid bases (e.g. line 235: PhytoSo)
Author Response
We thank you for reviewing our revision, and we have used the latest comments to do some further revision.
Comment 1: Figure 1(B): abbreviations in the figure do not fit the text now (Sa, So, PhytoSo etc.)
Response:We have corrected Figure 1B. Sa to DS, So to S and PhytoSo to P.
Comment 2: from cover letter of authors: ““In our revision, we include: “Note: Following molecular remodeling; i.e., conversion to various sphingoid bases and amide-linked fatty acids, certain amounts of Cer are converted to glycosylated Cer (mainly glucosylCer) and sphingomyelin” in the last part of Section 3.2.2.”” – cannot find it in the uploaded version 2.
Response:We have included “Note: Following molecular remodeling; i.e., conversion to various sphingoid bases and amide-linked fatty acids, certain amounts of Cer are converted to glycosylated Cer (mainly glucosylCer), omega-O-acylglucosylCers and sphingomyelin”in the last part of Section 3.2.3.
Comment 3: I appreciate the explanation the authors provided about my comments, however some parts of these explanations should be still included in the manuscript.
Response: We have included that statement in the last part of the Discussion section
“Our study could become a basis for developing topical agents containing Cer. Since we now know the fate of exogenous Cer and the effect of exogenous Cer on KC differentiation, we could test the effects of exogenous Cer in an appropriate formulation (that enhances the penetration of Cer into the stratum corneum) on epidermal barrier function. using in vivo/ex vivohuman skin”
Comment 4: Discussion, line 354-360: “Our study suggests that topical Cer agents are classified into two groups. …” – do you suggest this classification for the first time or is it already established? Rephrase the sentence to make it clear. (e.g., Based on our study we propose following classification of topical Cer agents…).
Response: This classification has notbeen proposed by others. As suggested, we include a revised sentence.
Insights gained from our study propose that topical Cer agents are classified into two groups. Group 1 includes Cer agents that improve epidermal barrier on the skin surface and in the SC. Group 2 includes Cer agents that work in KCs in nucleated layers; i.e.,endogenous Cer synthesis increases viaincreases in Cer synthesis precursors and by promoting KC differentiation.
Comment 5: I would recommend checking the manuscript for typos and completeness of renaming of the sphingoid bases (e.g., line 235: PhytoSo)
Response:We have re-checked abbreviations of sphingoid bases and ceramides throughout the manuscript.